# Actigraphy assessment of motor activity and sleep in patients with alcohol withdrawal syndrome and the effects of intranasal oxytocin

**Katrine Melby**[1,2,3]ʘ*, **Ole B. Fasmer**[4,5]ʘ, **Tone E. Henriksen**[4,6]‡, **Rolf W. Gråwe**[7,8]‡, **Trond O. Aamo**[2,3]‡, **Olav Spigset**[1,3]ʘ

**1** Department of Clinical and Molecular Medicine, Faculty of Medicine and Health Sciences, Norwegian University of Science and Technology–NTNU, Trondheim, Norway, **2** Blue Cross Lade Addiction Treatment Center, Trondheim, Norway, **3** Department of Clinical Pharmacology, St. Olav University Hospital, Trondheim, Norway, **4** Department of Clinical Medicine, Section for Psychiatry, University of Bergen, Bergen, Norway, **5** Division of Psychiatry, Haukeland University Hospital, Bergen, Norway, **6** Division of Mental Health Care, Valen Hospital, Fonna Health Authority, Valen, Norway, **7** Department of Research and Development, Division of Psychiatry, St. Olav University Hospital, Trondheim, Norway, **8** Department of Mental Health, Faculty of Medicine and Health Sciences, Norwegian University of Science and Technology–NTNU, Trondheim, Norway

ʘ These authors contributed equally to this work.
‡ These authors also contributed equally to this work.
* Katrine.melby@stolav.no

**Data Availability Statement:** All relevant data are available in figshare: 10.6084/m9.figshare. 11309624.

## Abstract

### Background and aims

The alcohol withdrawal syndrome increases autonomic activation and stress in patients during detoxification, leading to alterations in motor activity and sleep irregularities. Intranasal oxytocin has been proposed as a possible treatment of acute alcohol withdrawal. The aim of the present study was to explore whether actigraphy could be used as a tool to register symptoms during alcohol detoxification, whether oxytocin affected actigraphy variables related to motor activity and sleep compared to placebo during detoxification, and whether actigraphy-recorded motor function during detoxification was different from that in healthy controls.

### Methods

This study was a part of a randomized, double blind, placebo-controlled trial in which 40 patients with alcohol use disorder admitted for acute detoxification were included. Of these, 20 received insufflations with intranasal oxytocin and 20 received placebo. Outcomes were actigraphy-recorded motor activity during 5-hour sequences following the insufflations and a full 24-hour period, as well as actigraphy-recorded sleep. Results were related to clinical variables of alcohol intake and withdrawal, including self-reported sleep. Finally, the actigraphy results were compared to those in a group of 34 healthy individuals.

**Funding:** : This trial is granted by The Department of research and Development – AFFU at St. Olavs University Hospital, and funded by the Faculty of Medicine and Health Sciences, NTNU, the Research Department at St. Olavs University Hospital and the Joint Research Council between St. Olavs University Hospital and the Faculty of Medicine and Health Sciences, NTNU (FFU), Trondheim, Norway. The funders had no role in study design, data collection and analysis, decision to publish, or preparation of the manuscript.

**Competing interests:** The authors have declared that no competing interests exist.

## Results

There were no significant differences between the oxytocin group and the placebo group for any of actigraphy variables registered. Neither were there any correlations between actigraphy-recorded motor function and clinical symptoms of alcohol withdrawal, but there was a significant association between self-reported and actigraphy-recorded sleep. Compared to healthy controls, motor activity during alcohol withdrawal was lower in the evenings and showed increased variability.

## Conclusion

Intranasal oxytocin did not affect actigraphy-recorded motor activity nor sleep in patients with acute alcohol withdrawal. There were no findings indicating that actigraphy can be used to evaluate the degree of withdrawal symptoms during detoxification. However, patients undergoing acute alcohol withdrawal had a motor activity pattern different from than in healthy controls.

## Introduction

Increased anxiety, changes in motor activity and sleep disturbances are some of the clinical features of alcohol withdrawal syndrome (AWS), reflecting a hyper-excitatory state in the sympathetic part of the central nervous system that develops within the first 24 hours after drinking cessation [1, 2]. During detoxification from alcohol, rest and treatment with benzodiazepines are recommended to reduce symptoms of AWS [3, 4].

Exogenously administered oxytocin has been investigated related to its calming and anxiolytic effects in both healthy volunteers and in patients with depression and anxiety disorders [5–10]. In addition to increase parasympathetic activity, oxytocin has also been shown to be favorable on improving total sleep time and self-reported sleep satisfaction [11] Neither serious adverse events nor dependency to oxytocin have been reported in clinical trials, and it is generally considered to be a safe drug with few adverse effects [12].

A growing number of trials are exploring intranasal oxytocin as a potential drug in the treatment of alcohol use disorder and substance abuse, yet so far with mixed results [13–17]. One pilot trial suggested that the anxiolytic effects of intranasal oxytocin may be favorable in reducing alcohol withdrawal symptoms [18]. However, in a larger study from our group no differences between the oxytocin and placebo group were found [19]. The complexity of the effects of oxytocin has not yet been fully understood, and difficulties in measuring these effects [20, 21] as well as the complexity of assessing alcohol withdrawal, might explain the discrepancy between the two studies.

Actigraphy is a non-invasive, recognized tool for measuring motor activity, rest and sleep [22, 23], and the use of actigraphy can provide a better understanding of the motor activity and sleep patterns in AWS [24]. To our knowledge, no previous studies have used actigraphy to assess changes in motor activity and sleep in this setting. We therefore aimed to explore whether actigraphy could be used as a tool to register clinical symptoms of AWS during a 3-day period alcohol detoxification. We also wanted to explore whether intranasal oxytocin given during detoxification would affect actigraphy-recorded motor activity and sleep and to compare actigraphy registrations in patients with AWS with those from a control group of healthy subjects.

## Materials and methods

This study consists of two parts: In part 1, we compared, within the frames of a double-blind randomized controlled trial published previously [19], the effect of intranasal oxytocin and placebo on actigraphy-registered motor activity and sleep, during a 3-day course of alcohol detoxification. In part 2, we compared the actigraphy-registered motor activity and sleep in the total sample of 40 patients included in the randomized controlled trial to a control group of 34 healthy individuals.

The randomized controlled trial was approved by the Regional Committee for Medical and Health Research Ethics in Mid Norway (2016/45) and the Norwegian Medicines Agency (SLV) (2015-004463-37) and is registered in clinicaltrials.gov (identifier NCT02903251). It was also approved by the User Council at the addiction treatment center. The use of healthy volunteers was approved by the Regional Committee for Medical and Health Research Ethics in Western Norway in a separate application (2011/1668). All patients and controls gave their informed written consent prior to inclusion in the study.

### Subjects

In total, 40 patients were included between October 10th 2016 and November 17th 2017, as they were admitted to Lade Addiction Treatment Center, Trondheim, Norway to undergo alcohol detoxification. Inclusion criteria were 18–65 years of age, living in the county of Trøndelag, Norway. Moreover, the patient should have had either a prior episode of alcohol withdrawal symptoms causing significant incapacitation (e.g., inability to work or perform normal activities) or at least one prior medical detoxification with withdrawal symptoms of a magnitude requiring sedative-hypnotic or anticonvulsant medication. The average alcohol consumption should be in the range of 8–30 standard drinks per day for at least two weeks prior to enrolment in the study. The exclusion criteria were: daily treatment with sedative-hypnotic medications such as benzodiazepines or benzodiazepine-like hypnotics; dependence on substances other than alcohol, nicotine or caffeine; inadequately treated, unstable and/or compromising somatic or psychiatric conditions; a body mass index $< 17 \text{ kg/m}^2$ or a history of anorexia nervosa or bulimia in the past two years; pregnancy; parturition or breast-feeding in the past six months; and the inability to read or understand Norwegian sufficiently well to complete the study questionnaires. In addition, patients were excluded if the alcohol breath test at admission was negative, and the time interval since the last alcohol intake was more than 15 hours. Twenty patients were randomized to the oxytocin group and 20 to the placebo group. All patients fulfilled the ICD-10 diagnosis of alcohol use disorder, dependence (F10.2) (Fig 1).

The healthy controls primarily consisted of employees at the Department of Psychiatry, Fonna Regional Health Authority and Stavanger University Hospital, Norway. None of the controls were diagnosed with an affective or anxiety disorders, dependency to any drugs, nor were any of them prescribed psychotropic drugs. Data were collected between March 13th 2012 and June 6th 2013. Baseline characteristics of the patients and healthy controls are presented in Table 1.

### Procedures

Immediately after admission, patients were evaluated with the Clinical Institute of Withdrawal Assessment–Alcohol revised (CIWA-Ar) evaluation scale [25]. Withdrawal treatment with the benzodiazepine oxazepam was initiated according to the CIWA-Ar score before further study-specific procedures were performed. After giving their informed consent, patients were randomized to receive either 24 IU oxytocin or placebo double-blinded intranasally twice daily,

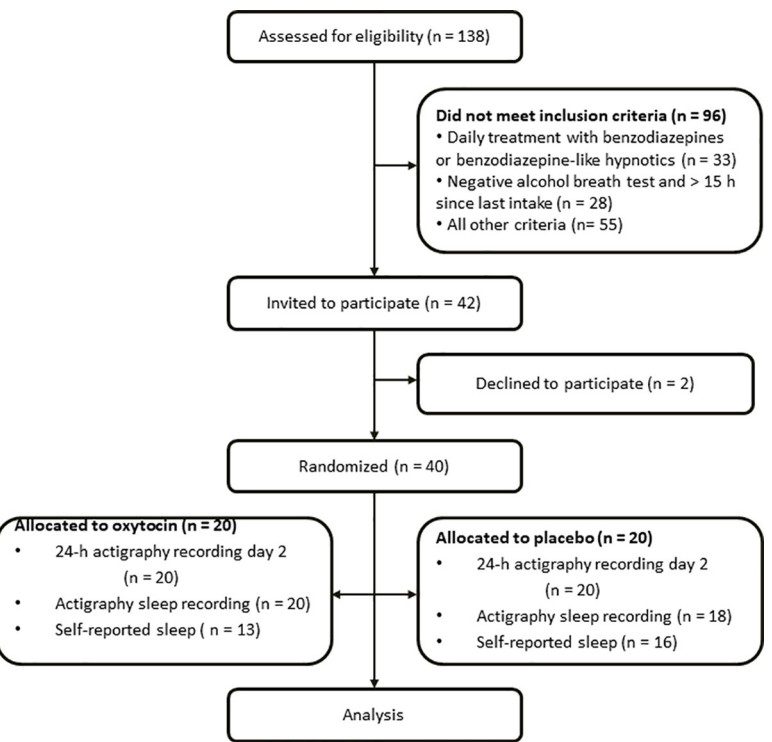

**Fig 1. Flowchart of patients with alcohol use disorder undergoing detoxification from alcohol included in the trial.**

and study nurses administered the first dose of nasal spray and applied the actigraph on the non-dominant wrist. The patients then followed the standard procedure for admission, including physical examination and repeated CIWA-Ar scorings according to the protocol. The CIWA-Ar is an assessment tool for alcohol withdrawal, and contains ten items, of which nine are scored from 1 to 7 and the last item from 1 to 4. High numbers are reflecting more severe withdrawal symptoms and a maximum score of 67 is possible. Treatment with benzodiazepines was commenced when the scores were 10 and above. Recorded actigraphy variables were related to total CIWA-Ar score as well as the three specific items anxiety, agitation and tremor.

**Table 1. Baseline characteristics of the 40 patients with alcohol use disorder undergoing detoxification from alcohol and the 34 healthy controls included in the study[1].**

| Variable | Patients | Healthy controls | p-value[2] |
|---|---|---|---|
| Number of subjects | 40 | 34 | - |
| Age (years), mean ± SD | 47.8 ± 10.4 | 42.0 ± 10.5 | 0.014 |
| Gender (males), n (%) | 28 (70.0) | 19 (55.9) | 0.209 |
| Employed, n (%) | 9 (22.5) | 32 (91.1) | <0.001 |
| Self-reported daily alcohol intake last 14 days (standard alcohol units[3]), mean ± SD | 16.0 ± 7.2 | NA | - |
| Phosphatidylethanol blood concentration, (μmol/L), mean ± SD | 2.23 ± 1.16[4] | NA | - |

NA = not available; SD = standard deviation

[1] Separate baseline data for those in the patient group treated with oxytocin (n = 20) and those treated with placebo (n = 20) are presented elsewhere [19].

[2] Student's t-tests for independent samples were used for continuous data, chi-square tests for categorical data.

[3] One standard alcohol unit corresponds to 12.8 g ethanol.

[4] N = 39

Patients were given a diary to register the number of hours asleep and the number of awakenings each of the two nights during the detoxification period. For all sleep measures, the objective assessment provided by the Actiware software was compared to patients' self-reported sleep, the registration of white light of the actigraph, and registered sleep in the patients' records when this information was available.

A Philips Respironics Actiwatch 2.0 was used in both the patient group and the healthy control group. Subjects were instructed in not remove it during the study period. In the patient group, data were sampled at 15-second intervals throughout the three study days. Data were retrieved and analyzed during a full 24-h period at day two as well as in morning and evening sequences throughout the study period. The 24-hour period lasted from 00:00 to 24:00. A morning sequence was defined as the time interval 09:00 to 14:00 and the evening sequence as 18:00 to 23:00. These intervals were chosen to correspond to the times for the administration of nasal spray and the expected duration of effect of oxytocin [26]. The healthy subjects wore the actigraph on the chosen wrist for a period of seven days. The time sequences used in the control group were the same as in the patient group.

## Data processing and statistics

The power calculation for the study sample was performed for the previously published study of effects of intranasal oxytocin on benzodiazepine consumption during alcohol withdrawal [19], where the main outcome was detection of a difference in oxazepam consumption between groups. A 10 mg difference in oxazepam was considered to be of clinical interest. Given an expected standard deviation of approximately 10 mg, power = 0.80 and alpha = 0.05, 16 subjects in each group was needed. Based upon an assumed drop-out rate of 20%, a total of 40 patients were included. The same patient sample was included in this study.

All data collected by the actigraph were analyzed using the following variables:

- The mean activity count per minute (mean/min), as a measure of overall activity level.

- The standard deviation for each time series in percent (SD/min in % of mean), as an intra-individual measure of fluctuations in activity.

- The root mean squared successive difference, describing the difference in successive counts from minute to minute (RMSSD/min in % of mean), as a measure of intra-individual variability in the time series.

- The RMSSD/SD ratio.

Motor activity was also analyzed based on the mean duration of the active and inactive periods (min), and also on the duration of the longest active period (min), the longest inactive period (min) and the ratio of mean duration of active/inactive periods.

Sleep was measured as self-reported sleep, sleep duration (all actigraphy registered sleep in evening and night), sleep efficiency (SE) (ratio between total sleep time and total duration of time in bed), and total sleep time (duration of sleep during the major sleep period in evening and night).

In part 1, Student's t-tests for independent samples were used to compare total activity and sleep between the two treatment groups. Motor activity was analyzed from the following pre-defined five-hour sequences twice daily during the three days of detoxification: The first five hours after admittance and inclusion, evening day 1, morning day, evening day 2, and morning day 3. For each sequence we analyzed the whole 300-minute period, and in addition the last 60-minute period with continuous motor activity in each sequence of the two treatment groups, and also the whole patient group. The criteria for a 60-minute period of continuously

motor activity followed those by Krane-Gartiser et al. [27], selecting the last 60-minute period not containing more than two consecutive minutes of zero activity counts from each participant. If that was not found, we selected 60-minute periods with between three and four consecutive minutes of zero activity. Paired samples t-tests were used to compare all consecutive morning and evening sequences to evaluate changes of total activity and in the last active 60 minutes in each period in the 40 patients.

In part 1, we also compared total activity and sleep during a 24-hour period at day 2 [28]. In addition to the measures listed above, each recorded interval (one minute) was defined as either active or inactive, according to whether the activity count was above or below 10% of the mean for the whole recording period for that participant. An active or inactive period was defined as a continuous sequence of such active or inactive intervals. Group differences of the mean duration of active periods (the longest active periods, the inactive periods, and the longest inactive periods) were compared with independent samples t-tests. Sleep variables used included sleep duration, sleep efficiency, and total sleep time [29, 30]. Difference in total duration of sleep in the first and the second night was compared with paired samples t-tests in the two treatment groups, as well as the mean sleep efficiency of both nights in the whole patient group. Over- or underestimation of sleep duration during the first two nights, defined as a deviation of > 10% from actigraphy recordings, was compared between the treatment groups. Pearson's test of correlations was used to evaluate the relationship between self-reported sleep, actigraphy-measured sleep, CIWA-Ar scores, and total oxazepam dose administered to the 40 patients in the 24-hour period, the first night and the second night.

In part 2, Student's t-tests for independent samples were used to compare the results from the patients recorded at day 2 with the healthy controls for total motor activity of the full 24-hour period, a five-hour morning sequence, and a five-hour evening sequence.

Statistical tests were performed with IBM SPSS Statistics for Windows, Version 25.0. [31] and the significance level was set to 0.05.

## Results

Of the 40 patients included, two patients in the placebo group had incomplete actigraphy recordings for some of the periods. One patient in each treatment group withdrew from the study, and did not complete day 3 (Fig 1). For the sleep analyses, 38 patients had complete actigraphy recordings, 20 in the oxytocin group and 18 in the placebo group. A total of 30 patients completed their sleep diary during the first and second nights and were included in the correlation analyses on self-reported and actigraphy measured sleep, withdrawal symptoms and craving. A typical actigram is shown in Fig 2.

### Part 1

There were no significant differences in daily starting times for actigraphy registration between the oxytocin group and the placebo group (13:51 and 13:13, respectively). The mean daily number of standard alcohol units consumed in the 14 days prior to detoxification was 17.0 ± 8.0 in the oxytocin group and 15.0 ± 6.4 in the placebo group (p = 0.399).

Consecutive comparisons of morning versus evening activities of the five sequences in the complete patient group are shown in Fig 3. Mean total and active period activities were generally significantly lower in the evening sequences than in the morning sequences. The opposite trend was found for SD and RMSSD. Comparison of the total motor activity during a 24-hour sequence showed no significant differences between the treatment groups (Table 2). Sleep did not differ significantly between the two treatment groups in any of the outcome variables in the first, nor in the second night (Table 3). Overall, patients in both groups had a mean sleep

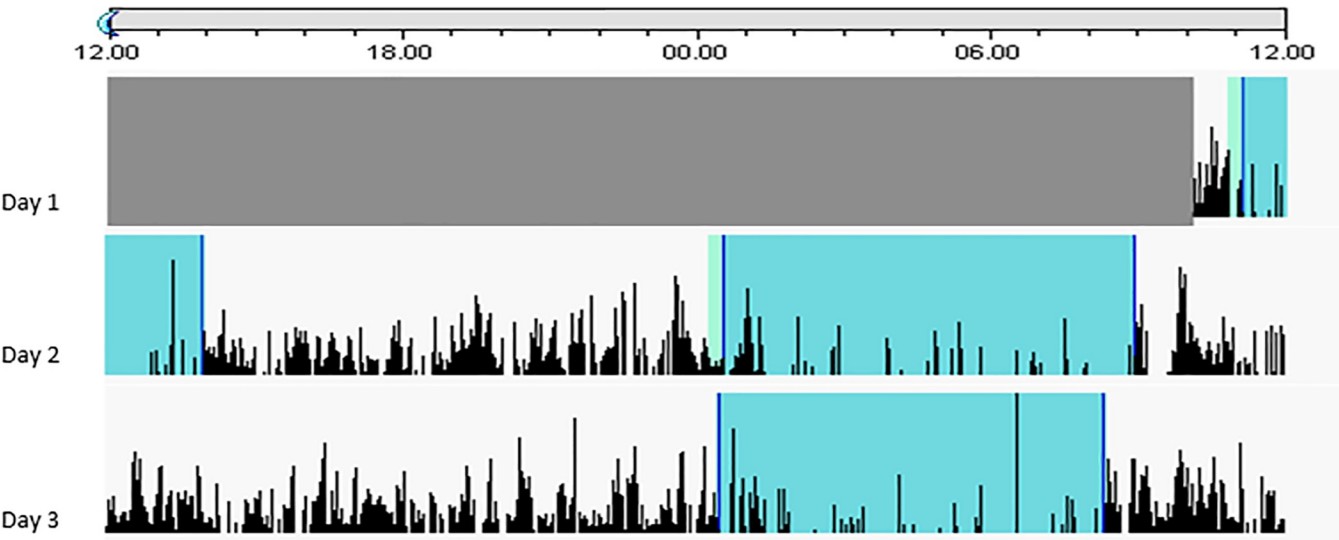

**Fig 2. Actigram showing motor activity and sleep during detoxification from alcohol.** The figure shows the actigram of one patient in the oxytocin group, with the activity counts in black, during three days of alcohol withdrawal. The height of a bar represents the motor activity during a time period. Grey areas indicate sleeping periods, as interpreted by the Actiware software.

efficiency of 79.1% (SD 13.9, range 38.1–97.9%) the first night, while the second night the mean sleep efficiency was 73.0% (SD 20.9, range 11.8–93.4%).

In general, there were no statistically significant correlations between clinical variables related to alcohol intake, withdrawal symptomatology, and actigraphy variables recorded in the 24-hour period on day 2 after admittance in the whole patient group (Table 4). The only exception was a significant correlation between the self-reported alcohol intake and RMSSD/ SD (r = 0.340; p = 0.032). Similar results were found when analyzing the oxytocin and the placebo groups separately (Panel A and B in S1 Table). The second night, irrespective of treatment with oxytocin or placebo, correlations between clinical variables related to alcohol intake and withdrawal and actigraphy-recorded sleep variables are presented in Table 5. There were significant correlations between self-reported sleep and actigraphy-recorded total sleep duration (r = 0.44; p = 0.017), sleep efficiency (r = 0.44; p = 0.017), and total sleep time (r = 0.44; p = 0.016), but no such clear-cut correlations were found within each treatment group (Panel A and B in S2 Table).

In total, 10 of the 29 patients from whom data were available estimated their sleep duration within ± 10% of the actigraphy-recorded sleep duration the first night, and 5 of 28 did the same the second night. The numbers of subjects overestimating and underestimating their sleep duration, as well as results from the placebo group and the oxytocin group shown separately, are presented in S3 Table. There was no obvious difference between the treatment groups in the number of subjects overestimating or underestimating their sleep duration, although the placebo group had a tendency of underestimating their sleep more than the oxytocin group did.

## Part 2

Motor activity of patients compared to healthy controls is summarized in Table 6. In the 24-hour period of day 2, the patient group showed significantly lower mean (±SD) activity (132 ± 57 vs. 198 ± 75 per minute; p < 0.001). This was also found in the evening sequence (174 ± 99 vs. 254 ±133 per minute; p = 0.005), but the difference did not reach statistical

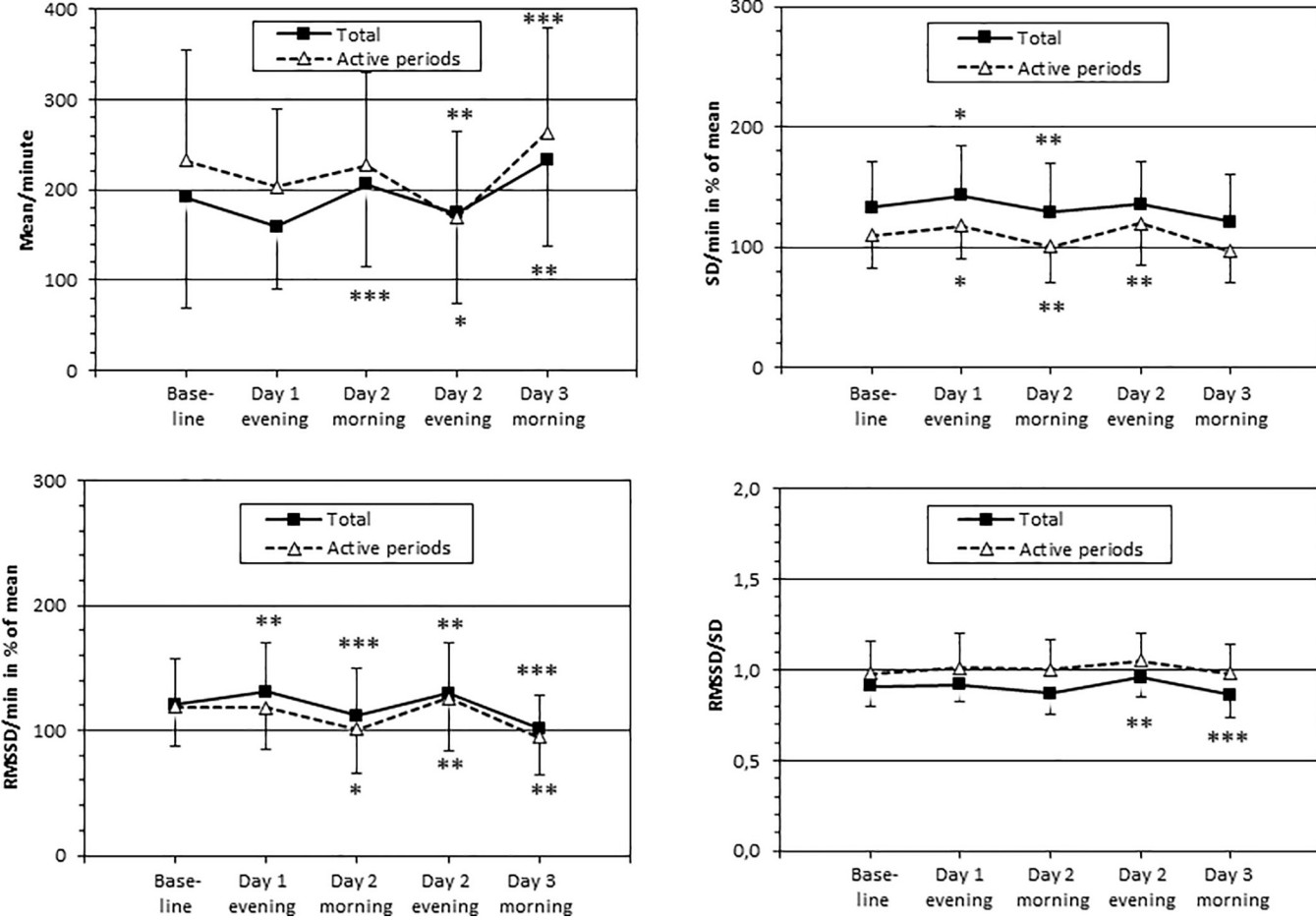

**Fig 3. Motor activity in 40 patients with alcohol use disorder during a 3-day course of detoxification.** Actigraphy recordings took place in 5-hour sequences in the mornings (09:00 to 14:00) and in the evenings (18:00 to 23:00). Values are presented as means ± standard deviations. Total motor activity and activity in the most active periods in each sequence are compared to those in the previous sequence using paired t-tests. * p < 0.05, ** p < 0.01, ***p < 0.001. (N = 39 Day 2 evening, N = 38 Day 3 morning).

significance in the morning sequence (206 ± 91 vs. 267 ±180 per minute; p = 0.083). The mean duration of active periods was significantly shorter in the patient group than the control group during the 24-hour period (6.0 ± 2.5 vs. 9.2 ± 3.5 min; p < 0.001). Moreover, motor activity in the periods with continuous activity was significantly lower in the patients than in the controls in the evening (169 ± 96 vs. 249 ±169 per minute; p = 0.020), but not in the morning (227 ± 101 vs. 263 ±197 per minute; p = 0.339). The patients also had increased total SD, RMSSD and RMSSD/SD (Table 6).

## Discussion

The principal finding in the present study is that there was no significant difference in actigraphy-recorded motor activity or sleep between the oxytocin and placebo groups for any of the variables in any of the sequences investigated. Moreover, there were no overall correlations between clinical variables related to alcohol intake or alcohol withdrawal and actigraphy-recorded motor activity and sleep. In contrast, patients had significantly lower motor activity than the healthy controls, particularly in the evening.

**Table 2. Comparisons between the oxytocin group (n = 20) and the placebo group (n = 20) for actigraphy-recorded motor activity and active and inactive periods during a 24-hour period and in a morning and an evening sequence in patients with alcohol use disorder undergoing detoxification from alcohol.**

| | Actigraphy variable | Oxytocin group (n = 20) | Placebo group (n = 20) | P value[1] | Difference (95% CI) |
|---|---|---|---|---|---|
| Total | Mean/min | 132 ± 58 | 133 ± 57 | 0.98 | -0.4 (-37.2, 36.3) |
| | SD/min in % of mean | 182 ± 43 | 173 ± 38 | 0.49 | 9.0 (-17.1, 35.1) |
| | RMSSD/min in % of mean | 144 ± 46 | 142 ± 45 | 0.90 | 1.8 (-27.2, 30.9) |
| | RMSSD/SD | 0.779 ± 0.091 | 0.812 ± 0.090 | 0.25 | -0.03 (-0.09, 0.02) |
| | Mean duration active period (min) | 6.4 ± 3.0 | 5.6 ± 1.8 | 0.34 | 0.8 (-0.8, 2.3) |
| | Duration longest active period (min) | 75.5 ± 48.7 | 73.2 ± 36.7 | 0.87 | 2.3 (-25.3, 29.9) |
| | Mean duration inactive period (min) | 6.2 ± 3.4 | 5.2 ± 1.6 | 0.24 | 1.0 (-0.7, 2.7) |
| | Duration longest inactive period (min) | 80.5 ± 110.5 | 54.3 ± 14.0 | 0.30 | 26.2 (-24.2, 76.6) |
| | Ratio, mean duration active/duration inactive period | 1.13 ± 0.50 | 1.18 ± 0.50 | 0.74 | -0.05 (-0.35, 0.25) |
| Morning sequence[2] | Mean/min | 205 ± 89 | 208 ± 95 | 0.92 | -3.1 (-62.1, 55.8) |
| | SD/min in % of mean | 132 ± 44 | 125 ± 39 | 0.56 | 7.7 (-62.1, 55.8) |
| | RMSSD/min in % of mean | 113 ± 38 | 110 ± 38 | 0.76 | 3.7 (-20.7, 28.1) |
| | RMSSD/SD | 0.868 ± 0.134 | 0.882 ± 0.118 | 0.72 | -0.01 (-0.10, 0,07) |
| Evening sequence[3] | Mean/min | 169 ± 72 | 179 ± 120 | 0.75 | -10.2(-75.1, 54.7) |
| | SD/min in % of mean | 138 ± 38 | 134 ± 33 | 0.2 | 4.2 (-19.0, 27.4) |
| | RMSSD/min in % of mean | 131 ± 43 | 130 ± 38 | 0.92 | 1.3 (-25.1, 27.6) |
| | RMSSD/SD | 0.943 ± 0.121 | 0.969 ± 0.103 | 0.47 | -0.03 (-0.10, 0.05) |
| Active period morning[4] | Mean/min | 211 ± 83 | 243 ± 116 | 0.33 | -32.1 (-97.5, 33.2) |
| | SD/min in % of mean | 106 ± 35 | 94 ± 25 | 0.24 | 11.6 (-8.2, 31.3) |
| | RMSSD/min in % of mean | 107 ± 36 | 94 ± 33 | 0.24 | 13.2 (-9.2, 35.7) |
| | RMSSD/SD | 1.024 ± 0.183 | 0.990 ± 0.165 | 0.55 | 0.34 (-0.08, 0.15) |
| Active period evening[5] | Mean/min | 170 ± 73 | 169 ± 117 | 0.98 | 0.869 (-63.1, 64.9) |
| | SD/min in % of mean | 120 ± 40 | 120 ± 32 | 1.00 | -0-05 (-23.6–23.5) |
| | RMSSD/min in % of mean | 127 ± 54 | 124 ± 29 | 0.83 | 3.0 (-25.4, 23.6) |
| | RMSSD/SD | 1.042 ± 0.144 | 1.052 ± 0.154 | 0.84 | -0.01 (-0.1, 0.09) |

All data are given as means ± standard deviations.

[1] Student's t-test for independent samples.

[2] 09:00 to 14:00

[3] 18:00 to 23:00. In total, 19 patients in the oxytocin group.

[4] The most active 60 minutes in the morning sequence. In total, 19 patients in the placebo group.

[5] The most active 60 minutes in the evening sequence. In total, 19 patients in the oxytocin group and 19 patients in the placebo group.

We originally had the idea that actigraphy could contribute to a better understanding of motor activity and sleep patterns during alcohol detoxification, and enable a more personalized intervention in acute alcohol withdrawal. However, there were no correlations between the total CIWA-Ar score or the degree of tremor, agitation and anxiety as measured by CIWA-Ar, and actigraphy-recorded motor activity. Investigating variables such as heart rate or skin conductance, which would have been of interest as they represent some of the most prominent clinical findings in acute alcohol withdrawal syndrome, would require another type of equipment. Thus, actigraphy, as least as used in the present study, seems to be a less appropriate method in the monitoring of acute alcohol withdrawal, although it might still be useful in the understanding of sleep disturbances during alcohol withdrawal.

Compared to healthy controls, the patients had a significantly lower motor activity, a finding that was prominent in the evening, but not in the morning. This could be a result of the patients' restrictions to leave the treatment facility in the evening. It is also possible that the alcohol withdrawal syndrome might have incapacitated them from being as active as they

**Table 3. Comparison of actigraphy-recorded sleep patterns between the oxytocin group (n = 20) and the placebo group (n = 18) in patients with alcohol use disorder undergoing a 3-day course of detoxification.**

| | | Oxytocin group (n = 20) | Placebo group (n = 18) | P value[3] | Difference (95% CI) |
|---|---|---|---|---|---|
| First night | Sleep duration (min) | 472 ± 118 | 477 ± 113 | 0.90 | -4.9 (-81.1, 71.2) |
| | Sleep efficiency (%)[1] | 80.6 ± 13.4 | 77.3 ± 14.7 | 0.47 | 3.3 (-5.9, 12.5) |
| | Wake after sleep onset (min) | 44.6 ± 24.6 | 43.3 ± 22.5 | 0.87 | 1.2 (-14.3, 16.8) |
| | Wake after sleep onset (%) | 9.9 ± 6.3 | 9.2 ± 5.2 | 0.73 | 0.7 (-3.2, 4.5) |
| | Total sleep time (min)[2] | 428 ± 120 | 434 ± 109 | 0.87 | -6.2 (-81.9, 69.5) |
| | Total sleep time (%) | 90.1 ± 6.3 | 90.8 ± 5.2 | 0.73 | -0.7 (-4.5, 3.2) |
| | Total activity counts | 7938 ± 4603 | 7735 ± 4824 | 0.90 | 203 (-2899, 3306) |
| | Activity counts per minute | 17.2 ± 10.3 | 14.7 ± 10.5 | 0.45 | 2.5 (-4.1, 9.2) |
| Second Night | Sleep duration (min) | 447 ± 148 | 452 ± 167 | 0.94 | -4.1 (-106.9, 98.8) |
| | Sleep efficiency (%)[1] | 71.8 ± 21.0 | 74.1 ± 21.4 | 0.74 | -2.3 (-16.0, 11.5) |
| | Wake after sleep onset (min) | 56.8 ± 34.2 | 51.0 ± 35.5 | 0.61 | 5.8 (-16.8, 28.4) |
| | Wake after sleep onset (%) | 12.7 ± 6.0 | 12.9 ± 9.8 | 0.96 | - 0.1 (-5.4, 5,1) |
| | Total sleep time (min)[2] | 391 ± 130 | 401 ± 162 | 0.84 | -9.9 (-105.2, 85.5) |
| | Total sleep time (%) | 87.3 ± 6.0 | 87.1 ± 9.8 | 0.96 | 0.1 (-5.1, 5.4) |
| | Total activity counts | 11574 ± 7452 | 9642 ± 9278 | 0.480 | 1932 (-3546, 7410) |
| | Activity counts per minute | 24.4 ± 13.7 | 24.0 ± 23.8 | 0.94 | 0.4 (-12.0, 12.9) |

All data are given as means ± standard deviations.

[1] Ratio between total sleep time and total duration of time in bed.

[2] Duration of sleep during the major sleep period in the evening/night.

[3] Student's t-test for independent samples.

otherwise might have been. Finally, the treatment with oxazepam could also have reduced their level of activity [27]. Longitudinal studies are required to elucidate these issues further.

Even though there was no difference between the oxytocin and placebo groups in activity in the morning and evening sequences, our findings are similar to the results in a study in affective disorders, where no differences in median activity in a 24-hour period were found between acutely admitted patients groups with unipolar depression, bipolar depression, mania or mixed states [28]. Patients undergoing acute alcohol withdrawal have a central nervous

**Table 4. Correlation between clinical variables related to alcohol intake and withdrawal, and actigraphy recordings of motor activity in 40 patients with alcohol use disorder in a 24-hour period on day 2 during detoxification from alcohol.**

| | Mean/min | | SD/min in % of mean | | RMSSD/min in % of mean | | RMSSD/SD | |
|---|---|---|---|---|---|---|---|---|
| | r | P value | r | P value | r | P value | r | P value |
| Self-reported daily alcohol intake last 14 days (standard alcohol units[1]), mean ± SD | -0.24 | 0.14 | -0.30 | 0.061 | -0.22 | 0.17 | **0.34** | **0.032** |
| Phosphatidylethanol blood concentration, (μmol/L), mean ± SD[2] | 0.09 | 0.58 | 0.01 | 0.95 | -0.19 | 0.91 | 0.04 | 0.81 |
| Total CIWA-Ar score | -0.06 | 0.73 | -0.05 | 0.76 | 0.02 | 0.89 | 0.23 | 0.16 |
| CIWA-Ar score, tremor | 0.17 | 0.29 | 0.03 | 0.87 | 0.08 | 0.62 | 0.10 | 0.56 |
| CIWA-Ar score, agitation | 0.18 | 0.27 | 0.11 | 0.48 | 0.09 | 0.59 | -0.06 | 0.73 |
| CIWA-Ar score, anxiety | 0.10 | 0.52 | 0.07 | 0.66 | 0.15 | 0.37 | 0.20 | 0.23 |
| Oxazepam dose | -0.02 | 0.91 | 0.004 | 0.98 | 0.07 | 0.68 | 0.15 | 0.36 |

CIWA-Ar = Clinical Institute of Withdrawal Assessment–Alcohol revised; SD = standard deviation. Statistically significant correlations are shown in bold.

[1] One standard alcohol unit corresponds to 12.8 g ethanol.

[2] N = 39

**Table 5. Correlation between clinical variables related to alcohol intake and withdrawal, and actigraphy recordings of sleep variables in 38 patients with alcohol use disorder in the second night during detoxification from alcohol.**

| | Sleep duration (min) | | Sleep efficiency[2] (%) | | Total sleep time[3] (min) | | Total sleep time (%) | |
|---|---|---|---|---|---|---|---|---|
| | r | P value | r | P value | r | P value | r | P value |
| Self-reported daily alcohol intake last 14 days (standard alcohol units[1]), mean ± SD | 0.29 | 0.078 | 0.18 | 0.28 | 0.24 | 0.15 | -0.07 | 0.67 |
| Phosphatidylethanol blood concentration, (μmol/L), mean ± SD | -0.03 | 0.85 | 0.05 | 0.79 | -0.08 | 0.64 | -0.09 | 0.60 |
| Total CIWA-Ar score | 0.08 | 0.61 | 0.23 | 0.17 | 0.11 | 0.52 | 0.10 | 0.55 |
| Oxazepam dose | 0.04 | 0.80 | 0.22 | 0.18 | 0.09 | 0.61 | 0.18 | 0.27 |
| Self-reported sleep | **0.44** | **0.017** | **0.44** | **0.017** | **0.44** | **0.016** | 0.18 | 0.35 |

CIWA-Ar = Clinical Institute of Withdrawal Assessment–Alcohol revised; SD = standard deviation. Statistically significant correlations are shown in bold.

[1] One standard alcohol unit corresponds to 12.8 g ethanol

[2] Ratio between total sleep time and total duration of time in bed

[3] Duration of sleep during the major sleep period in the evening/night

sympathetic hyper-excitation due to an alcohol-induced upregulation of glutamate receptors and the sudden loss of alcoholic GABA receptor stimulation [2]. This activation is also seen in patients with bipolar disorder in manic phase, and has been suggested to be a result of dysfunctional activation of the hypothalamic-pituitary-adrenal (HPA) axis and elevated noradrenalin and dopamine levels [32]; neurotransmitters also relevant in alcohol withdrawal. The overall increased variability seen in our study is in agreement with previous studies in bipolar disorder [27], depression [33] and attention deficit hyperactivity disorder (ADHD) [34]. Our findings do, however, contrast some of the results in another study where patients with bipolar disorder in manic phase had a significantly lower mean actigraphy-recorded activity than the healthy controls in the morning, and the only significant difference in the evening was a higher mean in successive counts from minute to minute (RMSSD/SD). Yet, this was not found in patients with depression [27].

Actigraphy is considered a practical and reliable method for monitoring sleep [22], and its validity has been confirmed when related to data from polysomnography [35, 36]. Yet, later studies have questioned whether it can be used to evaluate sleep disorders and circadian rhythm sleep-wake disorders [37] as well as its role for use in patients with insomnia and daytime symptoms [38].

Patients entering detoxification generally report sleep disturbances. In one study, all patients showed poor sleep efficiency (range of sleep efficiency 76–91%), as measured by actigraphy during acute withdrawal [24], a value previously measured to 94% in non-dependent participants [39]. The cut-off value for sleep efficiency has been recommended to be set to 92% [39]. In our study, poor sleep efficiency was also seen, with mean values of 72–81% in both treatment groups during the first and second nights (Table 3), despite symptomatic withdrawal treatment with oxazepam. Patients in both groups had a lower mean sleep efficiency the second night compared to the first night. This could be related to increasing symptoms of alcohol withdrawal the first two days of detoxification.

Somewhat surprisingly, the self-reported number of hours of sleep and actigraphy recorded sleep in both treatment groups correlated well. As many as 1/3 of the total patient group estimated their sleep duration within ± 10% of the actigraphy-recorded sleep duration the first night. This decreased to 1/5 the second night. Hence, actigraphy could be of importance for the accuracy of clinical sleep data in these patients in a longer period. We did not perform any further analyses on subgroups, e.g. based on PEth concentrations or self-reported alcohol intake, since the group sizes then would be too small.

**Table 6. Comparison of motor activity between 40 patients with alcohol use disorder and 34 healthy controls in a full 24-hour period as well as during the morning and evening sequences day 2 during detoxification from alcohol.**

| | Actigraphy variable | Patient group (n = 40) | Control group (n = 34) | P value[1] | Difference (95% CI) |
|---|---|---|---|---|---|
| Total | Mean/min | 132 ± 57 | 198 ± 75 | **<0.001** | -66.1 (-96.6, -35.5) |
| | SD/min in % of mean | 177 ± 41 | 145 ± 26 | **<0.001** | 32.1 (16.1, 47.7) |
| | RMSSD/min in % of mean | 143 ± 44 | 101 ± 21 | **<0.001** | 42.3 (26.5, 58.1) |
| | RMSSD/SD | 0.796 ± 0.091 | 0.692 ± 0.064 | **<0.001** | 0.10 (0.07, 0.14) |
| | Mean duration active period (min) | 6.0 ± 2.5 | 9.2 ± 3.5 | **< 0.001** | -3.2 (-4.6, -1.7) |
| | Duration longest active period (min) | 74.3 ± 42.6 | 126.5 ± 65.2 | **< 0.001** | -52.2 (-77.4, -27.1) |
| | Mean duration inactive period (min) | 5.7 ± 2.7 | 6.1 ± 1.9 | 0.474 | -0.4 (-1.5, 0.7) |
| | Duration longest inactive period (min) | 67.4 ± 78.8 | 62.6 ± 39.2 | 0.747 | 4.8 (-24.9, 34.5) |
| | Ratio, mean duration active/duration inactive period | 1.2 ± 0.5 | 1.5 ± 0.5 | **< 0.001** | -0.4 (-0.6, -0.2) |
| Morning sequence[2] | Mean/min | 206 ± 91 | 267 ± 180 | 0.083 | -60.3 (-128.8, 8.2) |
| | SD/min in % of mean | 129 ± 41 | 112 ± 50 | 0.126 | 16.5 (-4.7, 37.7) |
| | RMSSD/min in % of mean | 112 ± 38 | 91.0 ± 25.3 | **0.007** | 20.6 (5.8, 35.3) |
| | RMSSD/SD | 0.875 ± 0.125 | 0.855 ± 0.138 | 0.526 | 0.02 (-0.04, 0.08) |
| Evening sequence[3] | Mean/min | 174 ± 99 | 254 ± 132 | **0.005** | -79.4 (-133.6, -25.3) |
| | SD/min in % of mean | 136 ± 35 | 117 ± 40 | **0.041** | 18.3 (0.8, 35.8) |
| | RMSSD/min in % of mean | 130 ± 40 | 92.8 ± 30.9 | **<0.001** | 37.5 (20.6, 54.4) |
| | RMSSD/SD | 0.957 ± 0.112 | 0.798 ± 0.108 | **<0.001** | 0.16 (0.1, 0.2) |
| Active period morning[4] | Mean/min | 227 ± 101 | 263 ± 197 | 0.339 | -36.6 (-118, 38.5) |
| | SD/min in % of mean | 100 ± 31 | 94.0 ± 33.0 | 0.462 | 5.6 (-9.4, 20.5) |
| | RMSSD/min in % of mean | 100 ± 35 | 94.5 ± 30.6 | 0.418 | 6.3 (-9.2, 21,9) |
| | RMSSD/SD | 1.007 ± 0.173 | 1.018 ± 0.191 | 0.794 | -0.01 (-0.10, 0.08) |
| Active period evening[5] | Mean/min | 169 ± 96 | 249 ± 169 | **0.020** | -79.4 (-145.6, 15.6) |
| | SD/min in % of mean | 121 ± 35 | 105 ± 25 | **0.043** | 15.0 (0.5, 29.5) |
| | RMSSD/min in % of mean | 126 ± 43 | 101 ± 38 | **0.010** | 25.1 (6.1, 44.1) |
| | RMSSD/SD | 1.047 ± 0.147 | 0.944 ± 0.223 | **0.027** | 0.1 (0.01, 0.2) |

[1]Student's t-test for independent samples. All data are given as means ± standard deviations. Statistically significant p values are shown in bold.

[2]09:00 to 14:00

[3] 18:00 to 23:00. In total, 39 patients, 34 healthy controls.

[4] The most active 60 minutes in the morning sequence. In total, 39 patients, 33 healthy controls.

[5] The most active 60 minutes in the evening sequence. In total, 38 patients, 34 healthy controls.

This study has some limitations, but also some strengths. One of the strengths is that all 40 patients completed the 24-hour actigraphy recording, and as many as 38 of the 40 completed the full 3-day recording. Moreover, we have used the same methods and analyses as previous actigraphy studies on motor activity in psychiatric disorders, making the data directly comparable [27, 28].

No specific power calculation was performed for this study. The primary power calculation for the size of the clinical sample included in this study was based on the principal outcome in the main study published previously [19], to reveal a clinically relevant difference in oxazepam dose required between the oxytocin group and the placebo group during a three-day period of alcohol detoxification. In contrast, the present study had a pure exploratory aim, to evaluate whether actigraphy could be used as a tool to register symptoms of alcohol withdrawal syndrome during detoxification. Therefore, we consider it not relevant to perform a secondary power analysis for this study. Although it would have been possible to perform a separate power analysis for the comparison with the healthy volunteers, no previous information was available to determine the input data needed for a reliable analysis.

Among the limitations are that the five-hour sequences chosen to detect any changes in activity following the oxytocin nasal spray administration [12] might have caused some interesting activity and sleep data outside these periods to be lost. However, we consider this limitation to be accounted for by analyzing a full 24-hour period day two. We did not examine associations with caffeine use during detoxification, as these data were not available. Caffeine affects sleep in general [40, 41], although conflicting evidence exists on its influence on sleep during alcohol detoxification [24].

We did not investigate self-reported sleep or state of mood in the period prior to detoxification; therefore, we could not evaluate whether there was a sudden change in sleep or mood pattern. A previous trial on alcohol withdrawal and sleep in institutional detoxification [24] showed that poor sleep quality and insomnia were common in these patients, also prior to admission. The authors of that article discussed whether these symptoms were caused by a high alcohol intake or previous anxiety and/or depression, or if the mood alterations could be caused by insomnia [24].

Finally, it may be a weakness in the present study that we do not have any information on whether the dominant versus non-dominant wrist was chosen for actigraphy in the healthy controls. However, as discussed in previous trials[42], this factor should not be of importance for the principal results.

## Conclusion

Intranasal oxytocin did not affect actigraphy-recorded motor activity or actigraphy-derived sleep parameters in patients with acute alcohol withdrawal. Actigraphy does not seem to add information that can be used to evaluate the degree of withdrawal symptoms in the acute phase of alcohol detoxification, but further research is warranted to investigate whether in can provide useful information e.g. about sleep disturbances in the acute and protracted withdrawal period. During alcohol detoxification, patients have a motor activity pattern clearly different from that in healthy controls.

## Supporting information

**S1 Checklist. CONSORT 2010 checklist of information to include when reporting a randomised trial**\*.
(DOC)

**S1 Table. Correlation between clinical variables related to alcohol intake and withdrawal, and actigraphy recordings of motor activity in patients with alcohol use disorder in a 24-hour period on day 2 during detoxification from alcohol.** Panel A. In 20 patients receiving placebo. Panel B. In 20 patients receiving oxytocin.
(DOCX)

**S2 Table. Correlation between clinical variables related to alcohol intake and withdrawal, and actigraphy recordings of sleep variables in patients with alcohol use disorder in the second night during detoxification from alcohol.** Panel A. In 20 patients receiving placebo. Panel B. In 20 patients receiving oxytocin.
(DOCX)

**S3 Table. Number of subjects with alcohol use disorder overestimating, accurately estimating and underestimating their sleep duration in relation to actigraphy-recorded sleep during the first and second nights of acute detoxification.**
(DOCX)

**S1 Protocol.**
(DOCX)

**S2 Protocol.**
(DOCX)

## Acknowledgments

The authors would like to thank all participants in the study, all personnel at the Clinic of Addiction and Substance Abuse, St. Olav University Hospital and Blue Cross Lade Addiction Treatment Center, and in particular research nurses Linn Wigum, Mette Lykke Warholm and Bente Harsvik.

## Author Contributions

**Conceptualization:** Katrine Melby, Rolf W. Gråwe, Trond O. Aamo, Olav Spigset.

**Data curation:** Katrine Melby, Ole B. Fasmer, Tone E. Henriksen, Olav Spigset.

**Formal analysis:** Ole B. Fasmer, Tone E. Henriksen, Olav Spigset.

**Investigation:** Katrine Melby.

**Methodology:** Katrine Melby, Rolf W. Gråwe, Trond O. Aamo, Olav Spigset.

**Project administration:** Katrine Melby, Rolf W. Gråwe.

**Software:** Ole B. Fasmer.

**Supervision:** Rolf W. Gråwe, Trond O. Aamo, Olav Spigset.

**Visualization:** Katrine Melby, Olav Spigset.

**Writing – original draft:** Katrine Melby, Olav Spigset.

**Writing – review & editing:** Ole B. Fasmer, Tone E. Henriksen, Rolf W. Gråwe, Trond O. Aamo, Olav Spigset.

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
