## [Decision Letter · Decision Letter 0]

27 Oct 2019

PONE-D-19-23454

Actigraphy assessment of motor activity and sleep in patients with alcohol withdrawal syndrome and the effects of intranasal oxytocin

PLOS ONE

Dear Ms Melby,

Thank you for submitting your manuscript to PLOS ONE. After careful consideration, we feel that it has merit but does not fully meet PLOS ONE’s publication criteria as it currently stands. Therefore, we invite you to submit a revised version of the manuscript that addresses the points raised during the review process.

The manuscript has been assessed by two reviewers, their comments are available below.

The reviewers have raised some major concerns which need attention in a revision. The reviewers feel that the rationale for investigating sleep in this population should be further articulated and justified, and they recommend further discussion of how the study relates to earlier published work reporting the use of oxytocin on alcohol withdrawal symptoms. The reviewers note that the inclusion and exclusion criteria should be reported in greater detail and they raise questions about the sample size calculation.

Please carefully revise the manuscript to address the reviewers’ concerns. 

We would appreciate receiving your revised manuscript by Dec 09 2019 11:59PM. Please include the following items when submitting your revised manuscript:

We look forward to receiving your revised manuscript.

Kind regards,

Iratxe Puebla

Senior Managing Editor, PLOS ONE

Journal Requirements:

3. Thank you for including your ethics statement: The randomized controlled trial was approved by the Regional Ethics Committee (2016/45) and the Norwegian Medical Agency (2015-004463-37) and is registered in clinicaltrials.gov (identifier NCT02903251). It was also approved by the User Council at the addiction treatment center. The use of healthy individuals was approved by the Regional Ethics Committee in a separate application (2011/1668). All patients and controls gave their informed written consent prior to inclusion in the study.

4. Thank you for submitting your manuscript for consideration by PLOS ONE.

We note that your study is closely related to the following publications, on which you are an author:

https://doi.org/10.1016/j.drugalcdep.2019.01.003

Please cite and discuss the above study in the introduction and discussion sections of your manuscript, clarifying how the present work is related to the previously published paper.

Please note that our second publication criterion states that "If a submitted study replicates or is very similar to previous work, authors must provide a sound scientific rationale for the submitted work and clearly reference and discuss the existing literature. Submissions that replicate or are derivative of existing work will likely be rejected if authors do not provide adequate justification." http://www.plosone.org/static/publication.action#results.

Thank you for your attention to this request.

6.  We note that you have indicated that data from this study are available upon request. PLOS only allows data to be available upon request if there are legal or ethical restrictions on sharing data publicly. For information on unacceptable data access restrictions, please see http://journals.plos.org/plosone/s/data-availability#loc-unacceptable-data-access-restrictions.

Reviewers' comments:

Reviewer's Responses to Questions

**Comments to the Author**

1. Is the manuscript technically sound, and do the data support the conclusions?

Reviewer #1: Partly

Reviewer #2: Partly

2. Has the statistical analysis been performed appropriately and rigorously? 

Reviewer #1: Yes

Reviewer #2: No

3. Have the authors made all data underlying the findings in their manuscript fully available?

Reviewer #1: Yes

Reviewer #2: Yes

4. Is the manuscript presented in an intelligible fashion and written in standard English?

Reviewer #1: Yes

Reviewer #2: Yes

5. Review Comments to the Author

Reviewer #1: This study could potentially present valuable data that could add to the literature in a meaningful way, however, I think the authors need to refine their presentation of this data to create a more logical rationale for investigating sleep in alcohol withdrawal in the context of an OT clinical trial. I have 3 major comments on this manuscript:

1 What is the rationale for expecting OT to affect actigraphy and sleep related outcome measures?

2 A more rational approach would be for Part 1 to establish group differences (AUD vs controls) in actigraphy/ sleep outcomes. Then look for an effect of OT on these outcomes in AUD patients in withdrawal with a rationale to do so. That is, there any evidence that OT affects motor activity or sleep (there is some literature on this).

3 This seems a lost opportunity to report whether this study replicated a previous study on the effect of OT on alcohol withdrawal symptoms, craving and benzo requirement during withdrawal (Pedersen et al. 2013). If it does not have an effect on withdrawal symptoms then why should it have an effect on activity and sleep disturbances related to alcohol withdrawal.

Reviewer #2: The manuscript entitled 'Actigraphy assessment of motor activity and sleep in patients with alcohol withdrawal syndrome and the effects of intranasal oxytocin' with the aims to explore whether 1) actigraphy could be used as a tool to register symptoms during detoxification, 2) oxytocin affected actigraphy variables related to motor activity and sleep compared to placebo during detoxification, and 3) actigraphy-recorded motor function during detoxification was different from that in healthy controls.

This is quite an interesting study but the manuscript requires further improvement.

Comments

Abstract

Line 53, the sentence requires revision.

Materials and Methods

Line 96-103, all the details to be labelled using the word inclusion and exclusion criteria. There was no information on the inclusion and exclusion criteria for the controls. What are the characteristics of these subjects i.e. medical history, social history, occupation, psychological (stress, anxiety etc) etc. More information to be provided on type of occupation.

Table 1 requires cosmetic changes. At least one decimal point for the percentages. The symbol % for individual figures to be omitted since it was stated on the left column. There was no alcohol level measurement for the healthy controls. The p value - for the number of subjects to be omitted.

Line 130, for from1. 1 to be spaced from the word from.

Line 189 - 194, these sections to be included in their respective part in the earlier section in Page 8.

Line 195, proper citation for the SPSS including the publisher name to be stated.

There was no consideration of the attrition rates in the sample size calculation to derive the sample size.

Results

Statistical test(s) to be denoted in all the tables.

Line 199-200, the word mean to be stated.

Line 216 & Line 217, p values to be stated.

Table 2, decimal points to be standardized. For standardization, the word mean to be written in all variables.

Table 3, 2 subjects who were not included in the analysis to be denoted in the table footnote. For the title, n=38 to be stated rather than 40. Please ensure all the numbers stated in the figures and tables title are correct.

Line 243. phosphatidylethanol blood, Oxazepam dose to be stated.

Line 250-152, p value to be provided.

Table 4 and Table 5, results to be presented according to oxytocin group, placebo group and overall group.

Table 6, separate group (oxytocin and placebo group) to be presented.

Figure 2, 3, n to be stated.

References to conform with the journal format.

6. PLOS authors have the option to publish the peer review history of their article (what does this mean?). If published, this will include your full peer review and any attached files.

Reviewer #1: No

Reviewer #2: No

---

## [Author Response · Author response to Decision Letter 0]

3 Dec 2019

PONE-D-19-23454 resubmission: «Actigraphy assessment of motor activity and sleep in patients with alcohol withdrawal syndrome and the effects of intranasal oxytocin»

Dear Editor-in-chief,

We are pleased to resubmit for publication the revised version of our manuscript “Actigraphy assessment of motor activity and sleep in patients with alcohol withdrawal syndrome and the effects of intranasal oxytocin». We appreciate the comments from the two reviewers and the Senior Managing Editor. We have addressed each of the comments as outlined below, with our own response in italics. We have also uploaded one manuscript file where changes are highlighted in yellow, and one without highlighted changes.

We have done our best to adhere to your formatting checklist, including file naming. We have removed the phrase “data not shown”, and all data described are now either presented in detail in the manuscript or available as Supplementary tables. 

The full name of the Ethics Committee can be found in the manuscript and in the submission form. 

Regarding the availability of individual data from this study. We have now uploaded all data in figshare.com. The DOI will be available after published manuscript. The reason for making data available upon request is that even if data is anonymized, it still contains potentially identifying or sensitive patient information making the patient recognizable to himself/herself, co-patients at the ward, etc., given the small sample size of the trial, the specific time period the study was carried out, and the relatively small total population of Trøndelag county, Norway. 

No results from this manuscript has been published or is under consideration for publication elsewhere. The research protocol was presented as a poster at the The ASAM 48th Annual Conference, April 6-10, 2017, New Orleans, LA, USA and at the 30th ECNP Congress, September 2-5, 2017, Paris, France. 

Some other results from this RCT was previously published in Drug and Alcohol Dependence, DOI: 10.1016/j.drugalcdep.2019.01.003. However, this publication did not refer to actigraphy measures, motor activity or sleep, but merely describes the clinical and pharmacological effects of oxytocin in relation to placebo. Actigraphy is a separate field of research, we have included a control group, the research group consists of other people, etc. The results from the current study are therefore best presented as an independent paper. 

There are no changes related to the financial disclosure since the first submission.

Reviewer #1 – comments: 

1 What is the rationale for expecting OT to affect actigraphy and sleep related outcome measures?

To clarify this issue, changes have been made in Introduction, please cf. the first three paragraphs, which are rewritten. 

2 A more rational approach would be for Part 1 to establish group differences (AUD vs controls) in actigraphy/ sleep outcomes. Then look for an effect of OT on these outcomes in AUD patients in withdrawal with a rationale to do so. That is, there any evidence that OT affects motor activity or sleep (there is some literature on this).

We discussed this issue thoroughly before we submitted the first version of the paper, and we have now again considered the possibility of changing the order of the two parts. The reason why we did compare the two treatment groups first, was that the origin of this study was a double-blind RCT where two treatment groups, OT and placebo, were compared. Since no statistically significant differences were found between the groups in any of the outcomes (information that we had to present first, i.e. in Part 1), we could view the patient group as one homogenous group, thereby comparing all patients to healthy controls (Part 2). Doing it the other way around would in our opinion be less logical, because the reader would then not know whether it could be justified or not to compare the whole patient group with the healthy controls, i.e. whether the differences would be caused by the underlying condition or the drug treatment.

3 This seems a lost opportunity to report whether this study replicated a previous study on the effect of OT on alcohol withdrawal symptoms, craving and benzo requirement during withdrawal (Pedersen et al. 2013). If it does not have an effect on withdrawal symptoms then why should it have an effect on activity and sleep disturbances related to alcohol withdrawal.

We realize that we have not been able to clearly convey the rationale behind this study. The pilot study of Pedersen et al. from 2013 showed a remarkable effect in favour of oxytocin in alcohol withdrawal. Such an effect was not seen in our study where we tried to replicate these results. The clinical results from our study are already published (ref. 19 in the present article). We have now clarified this issue in paragraph 3 in the Introduction as well as in the very first sentence in the Materials and methods section. In the current study, we wanted to explore further whether actigraphy could give a more precise description of the degree of withdrawal symptoms. We also wanted to see whether actigraphy could give more information about the effects of oxytocin, effects that we were not able to reveal in the first study, as we had the idea that actigraphy provides a more accurate and objective measure to motor activity and sleep, than self-reported symptoms do. We have made numerous changes in the manuscript to better describe this rationale.

Reviewer #2 – comments: 

• Inclusion and exclusion criteria. There was no information on the inclusion and exclusion criteria for the controls. What are the characteristics of these subjects i.e. medical history, social history, occupation, psychological (stress, anxiety etc) etc. More information to be provided on type of occupation.

o The inclusion and exclusion criteria are now described. Moreover, table 1 now contains all relevant information available for the control group as well as the patients.

• Table 1 requires cosmetic changes. At least one decimal point for the percentages. The symbol % for individual figures to be omitted since it was stated on the left column. There was no alcohol level measurement for the healthy controls. The p value - for the number of subjects to be omitted.

o Table 1 is revised as suggested. 

• There was no consideration of the attrition rates in the sample size calculation to derive the sample size.

o The power calculation is now included in the Materials and methods section. Attrition rates can now be calculated based on the numbers given (32 patients requested, 40 included). In the Discussion section, we comment on the relevance of the power calculation performed in relation to the present study with focus on actigraphy data.

Results

• Table 2, decimal points to be standardized. For standardization, the word mean to be written in all variables.

o The numbers of decimals are now standardized in all Tables. The word “mean” is included where appropriate; we have in addition included in the legend to the table that all numbers are means ± SDs.

• Table 3, 2 subjects who were not included in the analysis to be denoted in the table footnote. For the title, n=38 to be stated rather than 40. Please ensure all the numbers stated in the figures and tables title are correct.

o The description of the numbers of patients are corrected as suggested. 

• Table 4 and Table 5, results to be presented according to oxytocin group, placebo group and overall group.

o For the information presented in Table 4, group-specific data are presented in S1 Table, panel A (placebo) and B (oxytocin). For the information presented in Table 5, group-specific data are presented in S2 Table, panel A (placebo) and B (oxytocin).

• Table 6, separate group (oxytocin and placebo group) to be presented.

o All separate group data are now included in Table 2. 

We appreciate the thorough assessment and comments of our manuscript from the reviewers and the Senior Managing Editor, and hope that you will find our revision satisfactorily. 

We look forward to hearing from you again.

---

## [Decision Letter · Decision Letter 1]

15 Jan 2020

PONE-D-19-23454R1

Actigraphy assessment of motor activity and sleep in patients with alcohol withdrawal syndrome and the effects of intranasal oxytocin

PLOS ONE

Dear Ms Melby,

Thank you for submitting your manuscript to PLOS ONE. After careful consideration, we feel that it has merit but does not fully meet PLOS ONE’s publication criteria as it currently stands. Therefore, we invite you to submit a revised version of the manuscript that addresses the points raised during the review process.

Please address minor comments.

We would appreciate receiving your revised manuscript by Feb 29 2020 11:59PM. To enhance the reproducibility of your results, we recommend that if applicable you deposit your laboratory protocols in protocols.io, where a protocol can be assigned its own identifier (DOI) such that it can be cited independently in the future. For instructions see: http://journals.plos.org/plosone/s/submission-guidelines#loc-laboratory-protocols

We look forward to receiving your revised manuscript.

Kind regards,

Shahrad Taheri

Academic Editor

PLOS ONE

Reviewers' comments:

Reviewer's Responses to Questions

**Comments to the Author**

1. If the authors have adequately addressed your comments raised in a previous round of review and you feel that this manuscript is now acceptable for publication, you may indicate that here to bypass the “Comments to the Author” section, enter your conflict of interest statement in the “Confidential to Editor” section, and submit your "Accept" recommendation.

Reviewer #2: All comments have been addressed

2. Is the manuscript technically sound, and do the data support the conclusions?

Reviewer #2: (No Response)

3. Has the statistical analysis been performed appropriately and rigorously? 

Reviewer #2: (No Response)

4. Have the authors made all data underlying the findings in their manuscript fully available?

Reviewer #2: (No Response)

5. Is the manuscript presented in an intelligible fashion and written in standard English?

Reviewer #2: (No Response)

6. Review Comments to the Author

Reviewer #2: Authors have put in great effort to address the comments.

Minor comment(s)

Line 175, the symbol > and < for the power and alpha to be replaced with symbol = and the sentence to be rephrased.

7. PLOS authors have the option to publish the peer review history of their article (what does this mean?). If published, this will include your full peer review and any attached files.

Reviewer #2: No

---

## [Author Response · Author response to Decision Letter 1]

17 Jan 2020

Editor-in-chief

PLOS ONE

 Trondheim, Norway, January 16th, 2020

PONE-D-19-23454R1 resubmission: «Actigraphy assessment of motor activity and sleep in patients with alcohol withdrawal syndrome and the effects of intranasal oxytocin»

Dear Editor-in-chief,

We are pleased to resubmit for publication the second revised version of our manuscript “Actigraphy assessment of motor activity and sleep in patients with alcohol withdrawal syndrome and the effects of intranasal oxytocin». We appreciate the minor comment from the reviewer. We have addressed the comment as outlined below and changed the revised manuscript accordingly. We have also uploaded one manuscript file where changes are made with track changes, and one without changes.

There are no changes related to the financial disclosure since the first submission.

Comment from Reviewer #2: 

Authors have put in great effort to address the comments.

Minor comment(s):

Line 175, the symbol > and < for the power and alpha to be replaced with symbol = and the sentence to be rephrased.

Reply:

We once again appreciate the thorough assessment of our manuscript from the reviewers and the Senior Managing Editor. We have rewritten the part describing the power analysis as suggested be the reviewer.

We hope that you will find our revision satisfactorily and look forward to hearing from you again.

Please address all correspondence to:

Katrine Melby, MD

Department of Clinical Pharmacology, 

St. Olav University Hospital

P.O. Box 3250, 

N-7006 Trondheim, Norway

Phone: + 47 92 88 66 39/ Fax: + 47 72 82 91 12

E-mail: katrine.melby@stolav.no

Yours sincerely,

Katrine Melby

---

## [Editor Report · Decision Letter 2]

23 Jan 2020

Actigraphy assessment of motor activity and sleep in patients with alcohol withdrawal syndrome and the effects of intranasal oxytocin

PONE-D-19-23454R2

Dear Dr. Melby,

We are pleased to inform you that your manuscript has been judged scientifically suitable for publication and will be formally accepted for publication once it complies with all outstanding technical requirements.

With kind regards,

Shahrad Taheri

Academic Editor

PLOS ONE
---

## [Editor Report · Acceptance letter]

3 Feb 2020

PONE-D-19-23454R2 

Actigraphy assessment of motor activity and sleep in patients with alcohol withdrawal syndrome and the effects of intranasal oxytocin 

Dear Dr. Melby:

I am pleased to inform you that your manuscript has been deemed suitable for publication in PLOS ONE. Congratulations! Your manuscript is now with our production department. 

With kind regards,

on behalf of

Dr. Shahrad Taheri 

Academic Editor

PLOS ONE